# ImputeINR: Enhancing Time Series Imputation with Adaptive Group-based Implicit Neural Representations

## Abstract

Time series data frequently exhibit the presence of missing values, rendering imputation a crucial process for downstream time series tasks and applications. However, existing imputation methods focus on discrete data points and are unable to effectively model sparse data, resulting in particularly poor performance for imputing substantial missing values. In this paper, we propose a novel approach, ImputeINR, for time series imputation by employing implicit neural representations (INR) to learn continuous functions for time series. ImputeINR leverages the merits of INR that the continuous functions are not coupled to sampling frequency and have infinite sampling frequency, allowing ImputeINR to generate fine-grained imputations even on extremely absent observed values. In addition, we introduce a multi-scale feature extraction module in ImputeINR architecture to capture patterns from different time scales, thereby effectively enhancing the fine-grained and global consistency of the imputation. To address the unique challenges of complex temporal patterns and multiple variables in time series, we design a specific form of INR continuous function that contains three additional components to learn trend, seasonal, and residual information separately. Furthermore, we innovatively propose an adaptive group-based framework to model complex residual information, where variables with similar distributions are modeled by the same group of multilayer perception layers to extract necessary correlation features. Since the number of groups and their output variables are determined by variable clustering, ImputeINR has the capacity of adapting to diverse datasets. Extensive experiments conducted on seven datasets with five ratios of missing values demonstrate the superior performance of ImputeINR, especially for high absent ratios in time series.

## 1 Introduction

Time series tasks, mainly including classification, anomaly detection, and forecasting, are vital across numerous domains, such as healthcare (Schaffer et al., 2021; Morid et al., 2023), industrial monitoring (Liu et al., 2020; Li et al., 2023), traffic flow (Cai et al., 2020; Ma et al., 2021), and human motion (Pérez-D'Arpino & Shah, 2015; Wang et al., 2017). However, real-world time series datasets suffer from missing values due to reasons like sensor malfunctions, data collection errors, or irregular reporting intervals. The missing information negatively impacts the inference of time series models, making imputation extremely necessary for downstream tasks.

Time series data imputation mainly meets three challenges: capturing temporal patterns, modeling cross-channel correlations, and dealing with absent observed information. Researchers have already attempted to address the first two challenges. Some early works (Cleveland et al., 1990; West, 1997) decompose time series to capture and model temporal patterns. Subsequent studies (Oreshkin et al., 2020; Wu et al., 2023; Liu et al., 2023c) have built upon the idea of decomposition, extracting trend and seasonal information separately. On the other hand, some deep learning based methods (Du et al., 2023; Wang et al., 2024) achieve significant imputation performance by mapping inputs from the data space to the feature space to learn cross-channel correlations. However, the existing imputation methods do not involve the cases of extremely absent observed values. Most works assume that the proportion of missing values requiring imputation does not exceed 50%, which

means that these methods still require a certain amount of known information. However, in real-world scenarios, the proportion of missing values is likely to be even higher. How to perform imputation based on extremely absent observed information remains a challenging task.

Recently, implicit neural representation (INR) has emerged as an effective method for continuously encoding diverse signals (Liu et al., 2023b; Molaei et al., 2023). It learns continuous functions from discrete data points, mapping coordinates to signal values. By representing complex structures in a compact form, INR is not coupled to sampling frequency anymore, which allows for multi-sampling frequency inputs enabling effective feature extraction even with absent observed samples. Additionally, as a continuous function, INR has infinite sampling frequency, which means it can be queried at any coordinate. This capacity for infinite sampling frequency interpolation sets it apart from other imputation methods, making it a promising approach for fine-grained imputation. However, directly applying INR to time series imputation is ineffective due to the unique complexities of time series.

In this paper, we propose a novel time series imputation approach, named ImputeINR, which can simultaneously address the three challenges mentioned above. First, we learn the INR continuous function, enabling modeling based on absent observed values and infinite sampling frequency for fine-grained interpolation. A multi-scale feature extraction module is incorporated to capture patterns and dependencies at various temporal scales, further achieving fine-grained imputation. Second, a novel form of INR continuous function is proposed for capturing complex temporal patterns and cross-channel correlation features. More specifically, the function is mainly decomposed into three components to learn trend, seasonal, and residual information separately. To further model the intricate residual components, we innovatively propose an adaptive group-based architecture. It is a multilayer perceptron (MLP) network composed of global layers and group layers. The former focuses on correlation information across all channels, while the latter emphasizes correlation information among variables with similar distributions. To enable our architecture to adapt to diverse datasets, we apply variable clustering to determine the number of groups and their outputs. Experimentally, ImputeINR achieves the state-of-the-art imputation performance on seven benchmarks under five ratios of missing values and the improvement becomes greater as the mask rate increases. The major contributions of this paper are summarized as follows:

- We propose ImputeINR, which learns INR continuous function to represent the continuous time series data. It leverages the sampling frequency-independent and infinite-sampling frequency capabilities of INR to achieve fine-grained imputation with absent observed data.

- We design an adaptive group-based architecture which is a part of the INR continuous function. It consists of global layers and group layers to learn correlation information across all variables and among variables with similar distributions, respectively. The number of groups and the output of each group are determined by variable clustering, allowing our architecture to adapt to diverse datasets. We use a transformer-based feed-forward network to predict INR parameters.

- To the best of our knowledge, ImputeINR is the first imputation approach to focus on the condition of the extremely absent observed data (i.e., mask rate is set to 70%/90%).

- Extensive experiments show that ImputeINR outperforms other baselines on seven datasets under five ratios of masked values. It achieves a 62.7% relative improvement compared to the second-best results. The improvement becomes greater as the mask rate increases. We also provide detailed ablation studies, robustness analysis, and visual analysis.

## 2 RELATED WORK

### 2.1 TIME SERIES IMPUTATION

The earliest time series imputation methods are based on the statistical properties of the data, using mean/median values or statistical models to fill in missing values, such as Simple-Mean/SimpleMedian (Fung, 2006) and ARIMA (Afrifa-Yamoah et al., 2020). In addition, machine learning methods learn patterns in the data, demonstrating greater adaptability and accuracy. Prominent implementations of these approaches include KNNI (Altman, 1992) and MICE (Van Buuren & Groothuis-Oorshoorn, 2011). Although these methods are simple and easy to interpret, their limitations lie in capturing the complex temporal and variable information inherent in time series

data. Recently, there has been widespread interest in using deep models to capture complex temporal patterns for imputation of missing values, due to their powerful representation capabilities. Common architectures include RNN-based methods (M-RNN (Yoon et al., 2018) and BRITS (Cao et al., 2018)) , CNN-based methods (TimesNet (Wu et al., 2023)), MLP-based methods (DLinear (Zeng et al., 2023), TimeMixer (Wang et al., 2024)) and transformer-based methods (SAITS (Du et al., 2023), FPT (Zhou et al., 2023), iTransformer (Liu et al., 2024), ImputeFormer (Nie et al., 2024)). However, the existing methods ignore the condition of extremely absent observed data and fail to impute missing values with absent known information.

## 2.2 IMPLICIT NEURAL REPRESENTATIONS

INR uses neural networks to model signals as continuous functions rather than explicitly representing them as discrete points. It captures complex high-dimensional patterns in data by learning a continuous mapping from input coordinates to output values. Various scenarios have seen successful applications, such as 2D image generation (Saragadam et al., 2022; Liu et al., 2023a; Zhang et al., 2024), 3D scene reconstruction (Yin et al., 2022; Liu et al., 2023b; Yang et al., 2024), and video representations (Mai & Liu, 2022; Zhao et al., 2023; Kwan et al., 2024). Since INR learns a continuous function, it is not coupled to the resolution, which implies that the memory needed to parameterize the signal does not depend on spatial resolution but rather increases with the complexity of the underlying signal. Also, INR has infinite resolution, which means it can be sampled at an arbitrary sampling frequency. Therefore, we leverage this characteristic of INR to perform time series imputation tasks. Sampling from the continuous function of INR enables fine-grained imputation even with extremely absent observed data. To learn the INR for target signal, there are mainly two typical strategies: gradient-based meta-learning methods (Lee et al., 2021; Liu et al., 2023a) and feed-forward hyper-network prediction methods (Chen & Wang, 2022; Zhang et al., 2024). In this work, we use a transformer-based feed-forward method to predict the INR for time series data since it can be easily adopted to an end-to-end imputation framework.

## 3 METHODOLOGY

### 3.1 PROBLEM FORMULATION

Denote time series data with $N$ variables and $T$ timestamps as $\mathbf{X} = \{\mathbf{x}_1, \mathbf{x}_2, \ldots, \mathbf{x}_N\} \in \mathbb{R}^{N \times T}$. The time series data $\mathbf{X}$ is incomplete and the mask rate is $r \in [0, 1]$. The corresponding binary mask matrix can be defined as $\mathbf{M} = \{m_{n,t}\} \in \{0, 1\}^{N \times T}$, where $m_{n,t} = 1$ if $x_{n,t}$ is observed, and $m_{n,t} = 0$ if $x_{n,t}$ is missing. The imputation task is to predict the missing values $\mathbf{X}_{\text{miss}}$ such that the predicted values $\hat{\mathbf{X}}$ satisfy $\hat{\mathbf{X}} = F_\theta(\mathbf{X}, \mathbf{M})$, where $F_\theta$ mentions the model with parameters $\theta$. The goal is to minimize the reconstruction error between the masked data and the imputed data:

$$\mathcal{L}(\hat{\mathbf{X}}, \mathbf{X}_{\text{gt}}) = \frac{1}{|\mathbf{M}_{\text{miss}}|} \sum_{t=1}^{N} \sum_{n=1}^{T} (1 - m_{n,t}) \cdot (\hat{x}_{n,t} - x_{n,t})^2, \tag{1}$$

where $|\mathbf{M}_{\text{miss}}|$ is the total number of missing values in $\mathbf{X}$ and $\mathbf{X}_{\text{gt}}$ is the ground truth.

### 3.2 METHOD OVERVIEW

The core idea of ImputeINR is to leverage the ability of INR to learn continuous functions and enable to query at any timestamp to achieve fine-grained interpolation. However, since time series data has inherently intricate temporal patterns and multi-variable properties, using a simple MLP as the INR continuous function to fit it is challenging. To address these issues, we design a novel form of INR continuous function specifically for time series data. This new form includes three components to capture the trend, seasonal, and residual information to deal with the unique temporal patterns. Furthermore, to enhance the ability of ImputeINR to model multi-variable data, we propose an adaptive group-based architecture to learn complicated residual information. Each group focuses on variables with similar distributions. And we use a clustering algorithm to determine the number of groups and the variables each group outputs. To further enhance the imputation capability of ImputeINR, we incorporate a multi-scale feature extraction module to capture information at different scales, thereby improving fine-grained imputation performance.

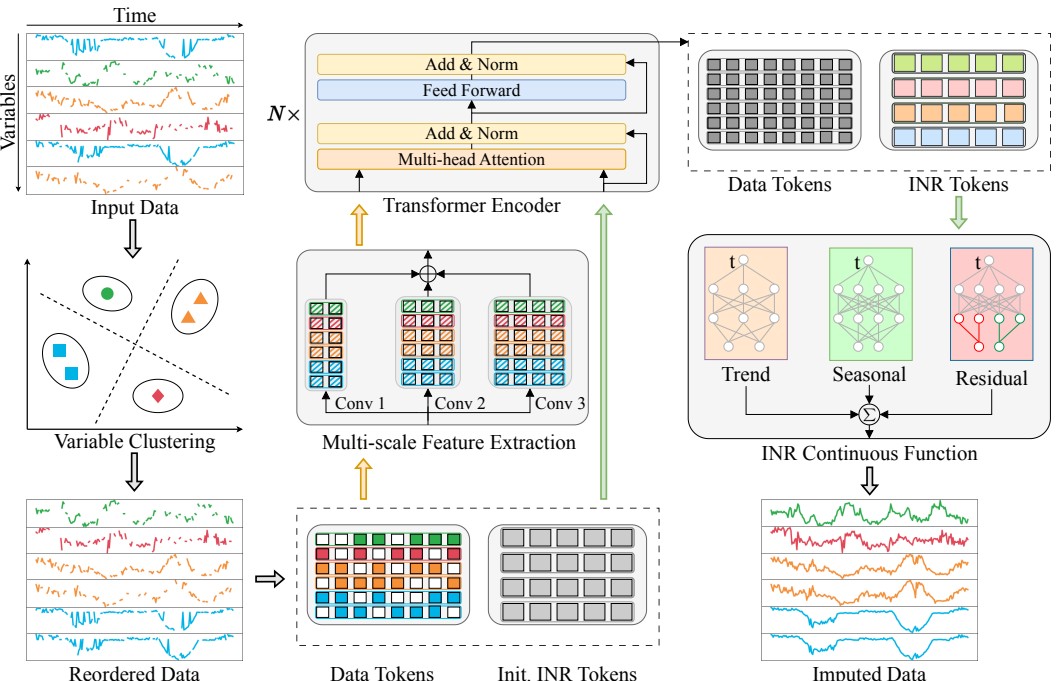

Figure 1: The overall workflow of the proposed ImputeINR method. The INR tokens are predicted using a transformer encoder. These tokens serve as the parameters for the INR continuous function, which takes the timestamp $t$ as input.

Figure 1 demonstrates the overall workflow of the proposed ImputeINR method. The masked data is first reordered based on the variable clustering results so that variables with similar distributions are placed adjacent to each other. This is to enable the subsequent representation of variables within the same cluster using the same group-based MLP in the INR continuous function. Then the reordered masked data is standardized and segmented into patches to prepare the data tokens. Simultaneously, we initialize the INR tokens, which are learnable vector parameters. The processed data tokens are input into convolutional layers of different scales to extract multi-scale features. Subsequently, these extracted features and the initialized INR tokens are fed together into the transformer encoder to predict the INR tokens. These INR tokens are essentially the parameters of the INR continuous function. Based on these parameters, INR continuous function takes the timestamp $t$ as input and predict the missing values.

## 3.3 VARIABLE CLUSTERING

We adopt a clustering algorithm $\mathcal{C}$ to cluster the variables of the time series data $\mathbf{X} \in \mathbb{R}^{N \times T}$ based on the similarity matrix $S \in \mathbb{R}^{N \times N}$, which partitions the variables into $K$ clusters:

$$\mathcal{C} : \mathbb{R}^{N \times N} \to \{C_1, C_2, \ldots, C_K\}, \tag{2}$$

where $C_k$ is a subset of the total variable set $\{\mathbf{x}_1, \mathbf{x}_2, \ldots, \mathbf{x}_N\}$ and its cardinality $|C_k|$ denotes the number of variables in this cluster. The objective of the clustering function $\mathcal{C}$ is defined as follows:

$$\mathrm{argmax}_{\{C_1, C_2, \ldots, C_K\}} \sum_{k=1}^{K} \sum_{\mathbf{x}_i, \mathbf{x}_j \in C_k} S\left(\mathbf{x}_i, \mathbf{x}_j\right), \tag{3}$$

where $S(\mathbf{x}_i, \mathbf{x}_j)$ represents the similarity between variables $\mathbf{x}_i$ and $\mathbf{x}_j$. Then we obtain the permutation matrix $P \in \mathbb{R}^{N \times N}$:

$$P_{ij} = \begin{cases} 1, & \text{if } j = \pi(i), \\ 0, & \text{otherwise}, \end{cases} \tag{4}$$

where $\pi$ is the permutation vector which orders the variables according to the clusters. Finally, the reordered matrix $\mathbf{X}'$ with columns permuted according to $\pi$ is given by:

$$\mathbf{X}' = \mathbf{X} \cdot P. \tag{5}$$

In this reordered matrix $\mathbf{X}'$, rows (i.e., variables) are grouped according to the clusters.

## 3.4 MULTI-SCALE FEATURE EXTRACTION

To further capture features from different scales for fine-grained imputation, the reordered data $\mathbf{X}' \in \mathbb{R}^{N \times T}$ is fed to multiple convolutional layers with varying scales. Each convolutional layer $l$ refers to kernel size $k_l$, stride $s_l$, padding $p_l$, and the number of output channels $c_l$. For each output channel $i$ in the $l^{th}$ convolutional layer, the convolution operation can be formulated as:

$$\Phi_l \left( \mathbf{X}' \right)_{i,t} = \sum_{j=1}^{k_l} W_{l,i,j} \cdot \mathbf{X}'_{t+j-p_l} + b_{l,i}, \tag{6}$$

where $W$ and $b$ denotes the weight matrix and bias matrix respectively. Then these features of different scales $\Phi_l \left( \mathbf{X}' \right) \in \mathbb{R}^{c_l \times (T - k_l + 2p_l + 1)}$ are concatenated to obtained the multi-scale features $\dot{\mathbf{X}} \in \mathbb{R}^{\sum_{l=1}^{L} c_l \times \left( T' - k_l + 2p_l + 1 \right)}$. Finally, these features are fed to the transformer encoder together with the initialized INR tokens to predict the INR tokens.

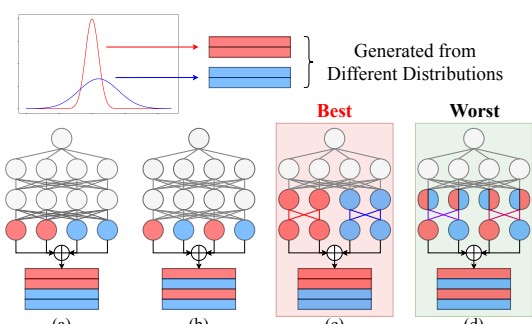

## 3.5 INR CONTINUOUS FUNCTION

INR continuous function $f$ maps the timestamp $t$ to time series data:

$$f : t \in \mathbb{R} \mapsto \mathbf{X}(t) \in \mathbb{R}^N, \tag{7}$$

where $\mathbf{X}(t)$ represents the output values of $N$ variables at timestamp $t$. To effectively capture the complicated temporal patterns and successfully model the multiple variables, we design a novel form of INR continuous function. Based on the idea of time series decomposition (Wen et al., 2019; Oreshkin et al., 2020), our INR continuous function includes three components

Figure 2: We manually synthesize a time series dataset with four variables, which are generated from two different distributions. Four different experimental settings are applied, including a single MLP with variables from the (a) same or (b) different distributions placed adjacent to each other, and a group-based MLP with variables from the (a) same or (b) different distribution in the same group. We observe that the setup in (c) performs the best, while (d) performs the worst.

to model trend, seasonal, and residual patterns separately. It can be defined as follows:

$$\hat{\mathbf{X}}(t) = f(t) = f_{tre}(t) + f_{sea}(t) + f_{res}(t), \tag{8}$$

where $t$ is the input timestamp and $f(t)$ denotes the output (i.e., imputed data). The parameters in INR continuous function are predicted by the transformer encoder (i.e., INR tokens).

**Trend Component** The trend represents the long-term movement or direction of the time series data, capturing the underlying pattern that shows whether the data is increasing or decreasing over time. It is typically smooth and reflects gradual shifts in the level of the time series, free from noise or short-term fluctuations. Mathematically, it can be modeled as a polynomial function:

$$f_{tre}(t) = \sum_{i=0}^{m} \alpha_i t^i, \tag{9}$$

where $\alpha_i$ denotes the coefficients and $m$ refers to the degree of the polynomial.

**Seasonal Component** The seasonal component focuses on the repeating patterns or cycles in the time series data, representing predictable fluctuations due to seasonality or recurring events. These regular, cyclical, and short-term fluctuations are modeled with a periodic function:

$$f_{sea}(t) = \sum_{i=1}^{\lfloor T/2 - 1 \rfloor} \left( \beta_i \sin \left( 2\pi i t \right) + \gamma_{i+\lfloor T/2 \rfloor} \cos \left( 2\pi i t \right) \right), \tag{10}$$

Table 1: Details of benchmark datasets.

| Dataset | Source | Dimension | Window | #Training | #Test |
|---|---|---|---|---|---|
| ETT | Electricity Transformer Temperature | 7 | 96 | 34465 | 11521 |
| Weather | Weather Station | 21 | 96 | 36696 | 10444 |
| Phy2012 | PhysioNet Challenge 2012 | 37 | 48 | 7671 | 2399 |
| Phy2019 | PhysioNet Challenge 2019 | 34 | 48 | 6104 | 1908 |
| BAQ | Beijing Multi-Site Air-Quality | 132 | 96 | 213 | 76 |
| IAQ | Italy Air Quality | 13 | 96 | 58 | 19 |
| Solar | Solar Alabama | 137 | 96 | 271 | 138 |

where $\beta_i$ and $\gamma_i$ are Fourier coefficients.

**Residual Component: Adaptive Group-based Architecture** The residual component represents the unexplained variation after removing the trend and seasonal effects, often modeled as a stochastic process. It is challenging to capture this complex information. As shown in Figure 2, we find that regardless of the order of the variables, using a single MLP is not effective in modeling multiple variables from different distributions. However, if variables from the same distribution are represented using the same set of MLP layers, the performance will significantly improve. We define such a set as a group. In addition, the layers in the MLP that extract information across all variables are called global layers, while the layers within groups are referred to as group layers. The number of groups and their outputs are determined by the results of variable clustering, which allows our architecture to adapt to datasets with various characteristics. It is worth noting that when variables with different distributions are in the same group, the representation capability is significantly reduced. This proves the importance of the correlation information between the variables. Detailed analysis can be found in Appendix A.3.

Theoretically, for any given timestamp $t$, we design $L_1$ global layers, $L_2$ group layers, and $K$ groups. $K$ is determined by the results of variable clustering. The global layers is given as follows:

$$h^{(0)} = t, \tag{11}$$

$$h^{(l_1)} = \sigma\left(W^{(l_1)} h^{(l_1-1)} + b^{(l_1)}\right), \tag{12}$$

where $l_1 \in [1, L_1]$, $h^{(l_1)}$ is the output of the $l_1^{th}$ global layer, $W$ and $b$ are weight matrix and bias matrix. Then, for group $g_k$, the input is the output of the last global layer:

$$y_{g_k}^{(0)} = h^{(L_1)}, \tag{13}$$

$$y_{g_k}^{(l_2)} = \sigma\left(W_{g_k}^{(l_2)} y_{g_k}^{(l_2-1)} + b_{g_k}^{(l_2)}\right), \tag{14}$$

where $l_2 \in [1, L_2]$, $y_{g_k}^{l_2}$ refers to the output of the $l_2^{th}$ group layer in group $g_k$, $W$ and $b$ are weight matrix and bias matrix. $y_{g_k}^{L_2} \in \mathbb{R}^{|C_k|}$ and $|C_k|$ is the number of variables in the $k^{th}$ cluster. The final output is the concatenation of the outputs from the last group layer of each group:

$$f_{res}(t) = y_{g_1}^{(L_2)} \oplus y_{g_2}^{(L_2)} \oplus \ldots \oplus y_{g_K}^{(L_2)}. \tag{15}$$

## 4 EXPERIMENTS

### 4.1 EXPERIMENTAL SETUP

**Datasets** We use seven time series imputation benchmark datasets to validate the performance of ImputeINR, including ETT (Zhou et al., 2021), Weather (Wetterstation), Phy2012 (Silva et al., 2012), Phy2019 (Reyna et al., 2019)), BAQ (Zhang et al., 2017), IAQ (Vito, 2016) and Solar (NREL, 2006). Table 1 shows details of the above benchmark datasets. These datasets are collected from different fields and have varying characteristics. Based on these datasets, we can evaluate the ability of models to handle varying numbers of variables and different sizes of datasets.

**Baseline Methods** We compare our proposed ImputeINR method to nine popular baselines, including statistical methods (Mean/Median), RNN-based methods (BRITS (Cao et al., 2018)), CNN-based methods (TimesNet (Wu et al., 2023)), MLP-based methods (TimeMixer (Wang et al., 2024)),

Table 2: Imputation results. The best results are in **Bold**. And the second ones are underlined.

| Methods / Mask Rate | | ImputeINR MSE | MAE | ImputeFormer MSE | MAE | TimeMixer MSE | MAE | iTransformer MSE | MAE | FPT MSE | MAE | TimesNet MSE | MAE | SAITS MSE | MAE | BRITS MSE | MAE | Transformer MSE | MAE | Mean/Median MSE | MAE |
|---|---|---|---|---|---|---|---|---|---|---|---|---|---|---|---|---|---|---|---|---|---|
| ETT | 10% | 0.020 | 0.098 | 0.021 | 0.091 | 0.035 | 0.115 | 0.042 | 0.141 | 0.017 | 0.087 | 0.018 | 0.088 | 0.021 | 0.100 | 0.021 | 0.089 | 0.021 | 0.099 | 1.104 | 0.790 |
| | 30% | 0.027 | 0.109 | 0.023 | 0.098 | 0.041 | 0.125 | 0.066 | 0.180 | 0.030 | 0.110 | 0.031 | 0.111 | 0.030 | 0.114 | 0.028 | 0.110 | 0.032 | 0.124 | 1.104 | 0.790 |
| | 50% | 0.028 | 0.111 | 0.034 | 0.116 | 0.054 | 0.143 | 0.109 | 0.234 | 0.041 | 0.130 | 0.035 | 0.123 | 0.031 | 0.116 | 0.040 | 0.130 | 0.044 | 0.147 | 1.104 | 0.790 |
| | 70% | 0.039 | 0.134 | 0.050 | 0.142 | 0.077 | 0.170 | 0.124 | 0.246 | 0.085 | 0.181 | 0.057 | 0.155 | 0.043 | 0.135 | 0.068 | 0.181 | 0.064 | 0.176 | 1.104 | 0.790 |
| | 90% | 0.095 | 0.214 | 0.122 | 0.218 | 0.223 | 0.276 | 0.247 | 0.336 | 0.272 | 0.309 | 0.231 | 0.295 | 0.213 | 0.218 | 0.251 | 0.358 | 0.234 | 0.335 | 1.104 | 0.790 |
| Weather | 10% | 0.026 | 0.063 | 0.032 | 0.076 | 0.029 | 0.069 | 0.036 | 0.081 | 0.028 | 0.064 | 0.028 | 0.064 | 0.031 | 0.073 | 0.027 | 0.063 | 0.030 | 0.080 | 0.634 | 0.606 |
| | 30% | 0.030 | 0.072 | 0.033 | 0.080 | 0.032 | 0.080 | 0.051 | 0.113 | 0.035 | 0.075 | 0.031 | 0.073 | 0.035 | 0.077 | 0.031 | 0.073 | 0.036 | 0.088 | 0.634 | 0.606 |
| | 50% | 0.031 | 0.073 | 0.037 | 0.084 | 0.037 | 0.076 | 0.069 | 0.144 | 0.043 | 0.076 | 0.036 | 0.076 | 0.041 | 0.091 | 0.035 | 0.077 | 0.042 | 0.097 | 0.634 | 0.606 |
| | 70% | 0.036 | 0.082 | 0.074 | 0.097 | 0.045 | 0.086 | 0.078 | 0.147 | 0.053 | 0.087 | 0.043 | 0.084 | 0.047 | 0.096 | 0.042 | 0.085 | 0.053 | 0.107 | 0.634 | 0.606 |
| | 90% | 0.065 | 0.123 | 0.082 | 0.116 | 0.076 | 0.126 | 0.124 | 0.191 | 0.089 | 0.129 | 0.073 | 0.125 | 0.066 | 0.124 | 0.090 | 0.130 | 0.099 | 0.173 | 0.634 | 0.606 |
| Phy2012 | 10% | 0.072 | 0.096 | 0.200 | 0.153 | 0.104 | 0.115 | 0.097 | 0.108 | 0.087 | 0.104 | 0.080 | 0.101 | 0.200 | 0.163 | 0.097 | 0.100 | 0.080 | 0.102 | 0.224 | 0.143 |
| | 30% | 0.079 | 0.101 | 0.205 | 0.155 | 0.117 | 0.120 | 0.099 | 0.111 | 0.099 | 0.111 | 0.103 | 0.108 | 0.203 | 0.168 | 0.108 | 0.105 | 0.094 | 0.107 | 0.224 | 0.143 |
| | 50% | 0.092 | 0.107 | 0.210 | 0.158 | 0.142 | 0.124 | 0.109 | 0.115 | 0.105 | 0.118 | 0.145 | 0.118 | 0.208 | 0.173 | 0.117 | 0.116 | 0.108 | 0.118 | 0.224 | 0.143 |
| | 70% | 0.071 | 0.112 | 0.229 | 0.169 | 0.148 | 0.129 | 0.124 | 0.120 | 0.132 | 0.131 | 0.149 | 0.128 | 0.237 | 0.195 | 0.125 | 0.123 | 0.122 | 0.125 | 0.224 | 0.143 |
| | 90% | 0.127 | 0.124 | 0.232 | 0.170 | 0.179 | 0.143 | 0.160 | 0.135 | 0.167 | 0.145 | 0.177 | 0.144 | 0.214 | 0.159 | 0.163 | 0.139 | 0.144 | 0.139 | 0.224 | 0.143 |
| Phy2019 | 10% | 0.071 | 0.102 | 0.199 | 0.159 | 0.100 | 0.116 | 0.072 | 0.104 | 0.082 | 0.111 | 0.075 | 0.105 | 0.199 | 0.168 | 0.089 | 0.103 | 0.080 | 0.105 | 0.203 | 0.153 |
| | 30% | 0.079 | 0.109 | 0.206 | 0.160 | 0.104 | 0.120 | 0.098 | 0.122 | 0.091 | 0.116 | 0.084 | 0.111 | 0.203 | 0.169 | 0.099 | 0.110 | 0.090 | 0.111 | 0.203 | 0.153 |
| | 50% | 0.087 | 0.115 | 0.209 | 0.164 | 0.109 | 0.125 | 0.100 | 0.123 | 0.102 | 0.124 | 0.094 | 0.118 | 0.204 | 0.175 | 0.109 | 0.118 | 0.099 | 0.119 | 0.203 | 0.153 |
| | 70% | 0.098 | 0.120 | 0.211 | 0.172 | 0.119 | 0.132 | 0.112 | 0.129 | 0.116 | 0.133 | 0.109 | 0.128 | 0.205 | 0.178 | 0.122 | 0.124 | 0.113 | 0.126 | 0.203 | 0.153 |
| | 90% | 0.121 | 0.131 | 0.214 | 0.174 | 0.152 | 0.149 | 0.123 | 0.132 | 0.153 | 0.152 | 0.149 | 0.149 | 0.206 | 0.180 | 0.151 | 0.142 | 0.137 | 0.137 | 0.203 | 0.153 |
| BAQ | 10% | 0.083 | 0.169 | 1.050 | 0.747 | 0.165 | 0.172 | 0.235 | 0.258 | 0.215 | 0.224 | 0.262 | 0.266 | 1.085 | 0.748 | 0.208 | 0.175 | 0.349 | 0.315 | 1.135 | 0.744 |
| | 30% | 0.096 | 0.171 | 1.096 | 0.749 | 0.205 | 0.193 | 0.308 | 0.321 | 0.231 | 0.229 | 0.292 | 0.267 | 1.088 | 0.749 | 0.210 | 0.186 | 0.387 | 0.324 | 1.135 | 0.744 |
| | 50% | 0.101 | 0.172 | 1.106 | 0.750 | 0.274 | 0.237 | 0.404 | 0.399 | 0.285 | 0.242 | 0.318 | 0.269 | 1.112 | 0.750 | 0.211 | 0.191 | 0.422 | 0.337 | 1.135 | 0.744 |
| | 70% | 0.117 | 0.181 | 1.119 | 0.751 | 0.359 | 0.289 | 0.556 | 0.488 | 0.325 | 0.262 | 0.341 | 0.280 | 1.124 | 0.751 | 0.230 | 0.206 | 0.448 | 0.359 | 1.135 | 0.744 |
| | 90% | 0.122 | 0.185 | 1.129 | 0.752 | 0.503 | 0.367 | 0.803 | 0.615 | 0.430 | 0.301 | 0.427 | 0.317 | 1.127 | 0.752 | 0.411 | 0.308 | 0.506 | 0.395 | 1.135 | 0.744 |
| IAQ | 10% | 0.007 | 0.061 | 1.340 | 0.725 | 0.139 | 0.171 | 0.592 | 0.466 | 0.228 | 0.264 | 0.248 | 0.286 | 1.277 | 0.735 | 0.164 | 0.210 | 0.599 | 0.514 | 1.493 | 0.767 |
| | 30% | 0.008 | 0.062 | 1.377 | 0.738 | 0.244 | 0.242 | 0.639 | 0.503 | 0.237 | 0.271 | 0.262 | 0.290 | 1.442 | 0.755 | 0.224 | 0.243 | 0.627 | 0.521 | 1.493 | 0.767 |
| | 50% | 0.009 | 0.063 | 1.424 | 0.753 | 0.375 | 0.306 | 0.783 | 0.556 | 0.291 | 0.305 | 0.274 | 0.297 | 1.461 | 0.757 | 0.241 | 0.273 | 0.710 | 0.553 | 1.493 | 0.767 |
| | 70% | 0.010 | 0.068 | 1.466 | 0.757 | 0.527 | 0.377 | 0.907 | 0.618 | 0.426 | 0.357 | 0.304 | 0.314 | 1.472 | 0.761 | 0.504 | 0.355 | 0.857 | 0.609 | 1.493 | 0.767 |
| | 90% | 0.029 | 0.116 | 1.478 | 0.761 | 0.847 | 0.498 | 1.205 | 0.767 | 0.811 | 0.504 | 0.720 | 0.477 | 1.493 | 0.764 | 0.981 | 0.505 | 1.201 | 0.716 | 1.493 | 0.767 |
| Solar | 10% | 0.022 | 0.074 | 0.768 | 0.771 | 0.024 | 0.079 | 0.060 | 0.167 | 0.075 | 0.173 | 0.048 | 0.132 | 0.770 | 0.772 | 0.023 | 0.075 | 0.061 | 0.128 | 0.773 | 0.775 |
| | 30% | 0.023 | 0.075 | 0.770 | 0.772 | 0.034 | 0.107 | 0.071 | 0.181 | 0.084 | 0.185 | 0.049 | 0.133 | 0.771 | 0.773 | 0.024 | 0.076 | 0.063 | 0.129 | 0.773 | 0.775 |
| | 50% | 0.024 | 0.078 | 0.772 | 0.773 | 0.052 | 0.143 | 0.079 | 0.189 | 0.101 | 0.202 | 0.052 | 0.139 | 0.772 | 0.774 | 0.026 | 0.080 | 0.065 | 0.135 | 0.773 | 0.775 |
| | 70% | 0.025 | 0.079 | 0.773 | 0.774 | 0.075 | 0.173 | 0.088 | 0.200 | 0.139 | 0.243 | 0.061 | 0.151 | 0.773 | 0.775 | 0.030 | 0.085 | 0.067 | 0.140 | 0.773 | 0.775 |
| | 90% | 0.026 | 0.081 | 0.774 | 0.775 | 0.166 | 0.249 | 0.120 | 0.250 | 0.435 | 0.444 | 0.121 | 0.211 | 0.774 | 0.776 | 0.052 | 0.100 | 0.077 | 0.158 | 0.773 | 0.775 |
| Average | 0.1 | 0.043 | 0.095 | 0.516 | 0.389 | 0.085 | 0.120 | 0.162 | 0.189 | 0.105 | 0.147 | 0.108 | 0.149 | 0.512 | 0.394 | 0.090 | 0.116 | 0.174 | 0.192 | 0.795 | 0.568 |
| | 0.3 | 0.049 | 0.100 | 0.530 | 0.393 | 0.111 | 0.141 | 0.190 | 0.219 | 0.115 | 0.157 | 0.122 | 0.156 | 0.539 | 0.401 | 0.103 | 0.129 | 0.190 | 0.201 | 0.795 | 0.568 |
| | 0.5 | 0.053 | 0.103 | 0.542 | 0.400 | 0.149 | 0.165 | 0.236 | 0.251 | 0.138 | 0.171 | 0.136 | 0.163 | 0.547 | 0.405 | 0.111 | 0.141 | 0.213 | 0.215 | 0.795 | 0.568 |
| | 0.7 | 0.057 | 0.111 | 0.560 | 0.409 | 0.193 | 0.194 | 0.284 | 0.278 | 0.182 | 0.199 | 0.152 | 0.177 | 0.557 | 0.413 | 0.160 | 0.166 | 0.246 | 0.235 | 0.795 | 0.568 |
| | 0.9 | 0.084 | 0.139 | 0.576 | 0.424 | 0.307 | 0.258 | 0.397 | 0.347 | 0.337 | 0.283 | 0.271 | 0.245 | 0.585 | 0.425 | 0.300 | 0.240 | 0.343 | 0.293 | 0.795 | 0.568 |

and transformer-based methods (Transformer (Vaswani, 2017), SAITS (Du et al., 2023), FPT (Zhou et al., 2023), iTransformer (Liu et al., 2024), ImputeFormer (Nie et al., 2024)). More details of these baselines are provided in Appendix A.4.

**Evaluation Metrics** We utilize Mean Square Error (MSE) and Mean Absolute Error (MAE) to report the imputation accuracy of all mentioned methods. These metrics are defined as follows:

$$\text{MSE} = \frac{1}{|\Omega|} \sum_{i,j \in \Omega} \left( \hat{\mathbf{X}}_{i,j} - \mathbf{X}_{\text{gt}_{i,j}} \right)^2, \text{MAE} = \frac{1}{|\Omega|} \sum_{i,j \in \Omega} \left| \hat{\mathbf{X}}_{i,j} - \mathbf{X}_{\text{gt}_{i,j}} \right|, \tag{16}$$

where $\mathbf{X}_{\text{gt}}$ is the ground truth, $\hat{\mathbf{X}}$ is the imputed data, $\Omega$ is the index set of masked entries.

**Experimental Settings** We apply the same data processing techniques and parameter settings. A sliding window approach is used, with a fixed window size of 48 for the Phy2012 and Phy2019 datasets, and 96 for all other datasets. These settings follow those used in previous work (Wu et al., 2023; Du, 2023). To evaluate the imputation performance, we randomly mask values in $\mathbf{X}_{\text{gt}}$ based on the mask rate $r$. For the main results, the multi-scale feature extraction module uses three parallel convolutional layers with kernel sizes of 3,5,7 respectively. The adaptive group-based architecture in the INR continuous function involves one global layer and one group layer within the residual component, with hidden dimensions set to 16. The transformer encoder consists of 6 blocks. Ablation Studies are reported in Section 4.3 to demonstrate the effectiveness of each module. Experiments are performed using the ADAM optimizer (Kingma, 2014) with an initial learning rate of $10^{-3}$. We use the agglomerative clustering method to achieve variable clustering since it adopts diverse inputs without the need to pre-specify the number of clusters. The visualization of the variable clustering results are provided in Section 4.5. All experiments are conducted on a single 24GB GeForce RTX 3090 GPU.

Table 3: The ablation studies on multi-scale feature extraction, variable clustering, and adaptive group-based architecture. Mask rate $r$ is 50% and the best results are in **Bold**.

| Multi-scale Features | Variable Clustering | Adaptive Group | ETT MSE | ETT MAE | Weather MSE | Weather MAE | Phy2012 MSE | Phy2012 MAE | Phy2019 MSE | Phy2019 MAE | BAQ MSE | BAQ MAE | IAQ MSE | IAQ MAE | Solar MSE | Solar MAE |
|---|---|---|---|---|---|---|---|---|---|---|---|---|---|---|---|---|
| ✗ | ✗ | ✗ | 0.039 | 0.135 | 0.038 | 0.083 | 0.099 | 0.114 | 0.098 | 0.119 | 0.227 | 0.262 | 0.018 | 0.092 | 0.036 | 0.106 |
| ✗ | ✗ | ✓ | 0.036 | 0.130 | 0.035 | 0.081 | 0.099 | 0.113 | 0.096 | 0.117 | 0.222 | 0.258 | 0.015 | 0.084 | 0.034 | 0.098 |
| ✗ | ✓ | ✗ | 0.036 | 0.129 | 0.036 | 0.080 | 0.098 | 0.113 | 0.097 | 0.118 | 0.218 | 0.259 | 0.015 | 0.083 | 0.033 | 0.096 |
| ✓ | ✗ | ✗ | 0.035 | 0.127 | 0.035 | 0.082 | 0.097 | 0.114 | 0.095 | 0.117 | 0.209 | 0.252 | 0.017 | 0.088 | 0.033 | 0.100 |
| ✗ | ✓ | ✓ | 0.029 | 0.115 | 0.032 | 0.074 | 0.093 | 0.108 | 0.088 | 0.111 | 0.192 | 0.243 | 0.010 | 0.066 | 0.031 | 0.092 |
| ✓ | ✗ | ✓ | 0.034 | 0.124 | 0.034 | 0.079 | 0.095 | 0.113 | 0.093 | 0.116 | 0.203 | 0.248 | 0.012 | 0.077 | 0.031 | 0.094 |
| ✓ | ✓ | ✗ | 0.033 | 0.123 | 0.033 | 0.078 | 0.096 | 0.113 | 0.094 | 0.117 | 0.199 | 0.244 | 0.014 | 0.081 | 0.032 | 0.096 |
| ✓ | ✓ | ✓ | **0.028** | **0.111** | **0.031** | **0.073** | **0.092** | **0.107** | **0.087** | **0.115** | **0.101** | **0.172** | **0.009** | **0.063** | **0.024** | **0.078** |

## 4.2 Main Results

We compare our ImputeINR method to nine popular baselines with five different mask rates $r$. As shown in Table 2, our ImputeINR achieves the best performance in most conditions in terms of both MSE and MAE metrics. Overall, across all datasets and mask rates, our method achieves an average MSE reduction of 62.7% compared to the second-best results. The superiority of ImputeINR is much more significant in IAQ, the dataset with fewest training samples. We observe similar improvements, occurring in other small datasets, BAQ and Solar. More specifically, the average MSE of our method is reduced by 16.6%, 54.9% and 96.1% on the Solar, BAQ and IAQ datasets respectively; while ImputeFormer and SAITS perform poorly on these datasets, yielding results similar to Mean/Median. These results demonstrate that our proposed ImputeINR can effectively deal with datasets of various sizes.

In addition, we observe that the performance of most methods declines as the mask rate $r$ increases. This aligns with our expectations, as fewer samples are captured leading to incomplete information, which increases the difficulty of imputation. However, ImputeINR is still effective even with an extreme mask rate. When 90% of the data is masked, the average MSE of our method is reduced by 69.2% compared to the second-best ones. This indicates that ImputeINR can learn continuous function from very few data points, achieving fine-grained imputation.

## 4.3 Ablation Studies

In this section, we conduct ablation studies to evaluate the effectiveness of multi-scale feature extraction block, variable clustering and adaptive group-based architecture. Table 3 presents the imputation results for all conditions. First, the model without any of the three modules exhibits the lowest performance. Building on this, adding any one of the modules will enhance the imputation capability of the model. This individually validates the effectiveness of each of the three modules. Furthermore, the permutation of any two modules will lead to higher performance. Among them, the combination of variable clustering and adaptive group-based architecture yields the best results. This is as expected, since the outcomes of variable clustering correspond directly to the number of groups. Therefore, these two modules can support each other, facilitating better representation learning. Finally, the model using all three modules displays the highest imputation performance.

## 4.4 Robustness Analysis

We further evaluate the robustness of our ImputeINR method on mask rate $r$ and the number of variables. As shown in Figure 3a, ImputeINR outperforms other comparison methods under all mask rate settings, proving its robustness on diverse missing ratio. Particularly, as the mask rate $r$ increases, the improvement of our method over others also becomes more pronounced. For example, when $r = 10\%$, the average MSE of our method is reduced by 49.5%, while at $r = 0.9$, the reduction reaches 69.2%. In addition, we also validate the robustness on the number of variables. As shown in Figure 3b, our method consistently performs the best on diverse numbers of variables. This demonstrates that our approach effectively addresses the challenges of multi-variable scenarios.

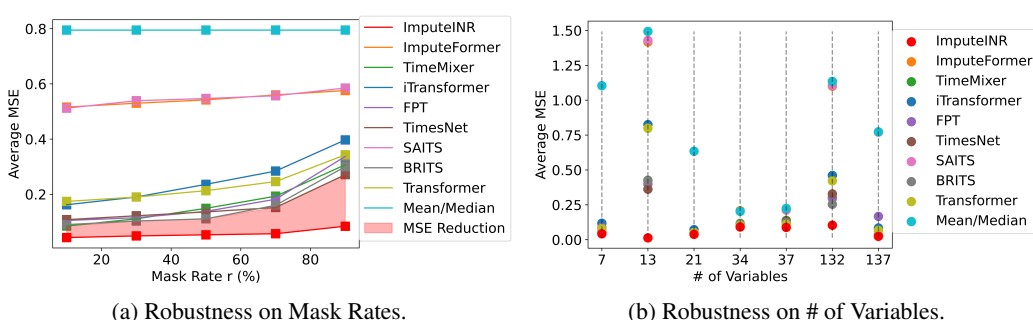

(a) Robustness on Mask Rates.  (b) Robustness on # of Variables.

Figure 3: Robustness analysis for mask rates and the number of variables.

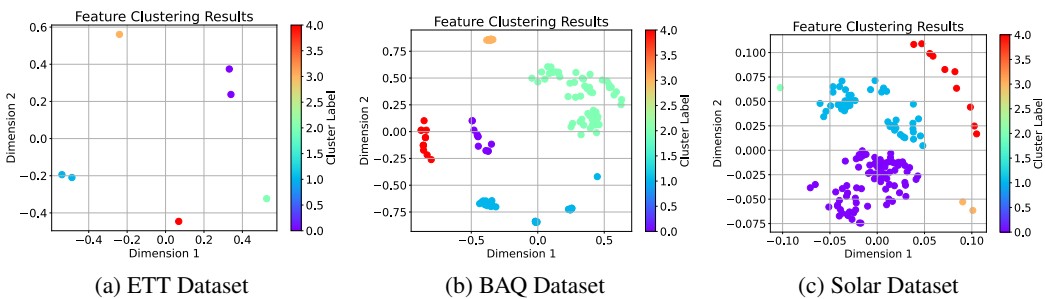

(a) ETT Dataset  (b) BAQ Dataset  (c) Solar Dataset

Figure 4: The visualization of variable clustering of (a) ETT dataset, (b) BAQ dataset, (c) IAQ dataset, and (c) Solar dataset. Variables with similar distributions are clustered and to be learned within the same group.

## 4.5 VISUAL ANALYSIS

In order to show the importance of variable clustering, we visualize the clustering results of ETT, BAQ, and Solar datasets. As shown in Figure 4, variables with similar distributions are clustered and will be assigned to the same group. We observe that even with a small number of variables, their distributions can vary significantly. Moreover, as the number of variables increases, the situation becomes more complex. In this case, using a single MLP to model all variables may weaken the unique local information within the same cluster, leading more focus on global information. Therefore, variable clustering is necessary, and previous ablation studies also prove this.

## 5 CONCLUSION

In this paper, we propose ImputeINR, an adaptive group-based time series imputation method. It learns the INR continuous function to map timestamps to the corresponding variable values. In contrast to existing imputation approaches, ImputeINR leverages the sampling frequency-independent and infinite-sampling frequency capabilities of INR to achieve fine-grained imputation with absent observed data. In addition, a multi-scale feature extraction module is added to further enhance fine-grained interpolation by capturing temporal patterns from different time scales. To model the complex information of time series data, we design a novel form of INR continuous function, which mainly includes three components to learn trend, seasonal, and residual information separately. Moreover, we propose an adaptive group-based architecture for the residual component. It focuses on correlation information across all variables and among variables with similar distributions through global layers and group layers respectively. We apply a variable clustering algorithm to determine the number of groups and the output dimension of each group, allowing the architecture to adapt to diverse datasets. Comprehensive experiments are conducted on seven imputation benchmark datasets under five ratios of masked values. The experimental results demonstrate that ImputeINR outperforms other state-of-the-art imputation methods. And the improvement becomes greater as less data is observed. In future work, we plan to explore the ability of INR for time series forecasting, which is the most challenging task.

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
