# A  APPENDIX

## A.1  DATA CONTINUITY AND IMPLICIT NEURAL REPRESENTATIONS

The real-world signals are not discrete, but they are represented in a discrete manner. For instance, we represent time series as sequences of discrete points, using sampled values at specific time intervals. However, these discrete representations come with a significant drawback: they only capture a absent amount of information about the signal. Therefore, to utilize these discrete sampling information to represent the complete continuous signal, we need to learn a continuous function $f$ that parameterizes the signal mathematically. With a timestamp $t$, $f$ outputs the corresponding values at that time. And we can sample the time series at any time point from $f$.

This type of continuous function is called implicit neural representation (INR). INRs are neural networks (e.g., MLPs) that estimate the function $f$ that represents a signal continuously by training on discretely represented samples of the same signal, based on the idea that neural networks can estimate complex functions after observing training data. The process to learn continuous function $f$ can be defined as follows:

$$f\left(x, \phi, \nabla_x \phi, \nabla_x^2 \phi, \cdots\right) = 0, \phi : x \mapsto \phi(x), \tag{17}$$

where $\phi$ is parameterized by the network and the estimated $f$ is implicitly encoded in the network after training on the discretely represented samples.

Unlike traditional representations that use discrete data points (like pixels in images or time points in time series), INRs encode information through functions that map coordinates (such as spatial positions or timestamps) directly to values (like colors or variables). Therefore, INRs allow for smooth representation of data, making it possible to generate high-resolution outputs from low-resolution inputs by querying the model at arbitrary points. And the continuous nature of INRs can lead to better generalization in tasks requiring interpolation or extrapolation of data.

## A.2  ALGORITHM

The algorithm of variable clustering (Algorithm 1) and the overall ImputeINR imputation method (Algorithm 2) are presented as follows. All experiments are implemented based on PyTorch.

For the variable clustering (Algorithm 1), we employ the agglomerative clustering method because it allows for varied inputs and does not require a predetermined number of clusters. Agglomerative clustering is a hierarchical clustering technique that starts with each data point as its own individual cluster. The algorithm iteratively merges the closest pairs of clusters based on a chosen distance metric until a stopping criterion is met. This method is particularly useful for its flexibility, as it does not require the number of clusters to be specified in advance, making it suitable for exploratory data analysis. With this clustering method, we obtain the clusters $C$ which is used to determine the specific settings of the adaptive group-based architecture.

For the overall ImputeINR imputation method (Algorithm 2), we predict the masked values with our designed INR continuous function. The masked data is firstly reordered based on the variable clustering results to make variables with similar distributions adjacent and then fed into a multi-scale feature extraction module to capture information from different time scales. The extracted features are entered in a transformer encoder with initialized INR tokens to predict the INR tokens. These learned INR tokens are the parameters of the INR continuous functions. More specifically, these parameters are not learnable but are predicted by the transformer encoder. With the predicted parameters, we input timestamp $t$ to calculate the corresponding variable values as the imputed data. The objective function (i.e., loss function) is the reconstruction error between the masked data and the imputed data as mentioned in Equation 1.

## A.3  REPRESENTATION CAPABILITY OF INR FOR TIME SERIES

To evaluate the representation capability of the INR continuous function for time series, we synthetically create a time series dataset and conduct several validation experiments. The synthetic dataset includes four variables, with two variables sampled from a normal distribution with a mean of 0 and a variance of 1, and the other two variables sampled from a normal distribution with a mean of 1 and a variance of 3. In other words, the four variables are generated from two different distributions.

---

**Algorithm 1** Variable Clustering

---

1: **Input:** Data points $\mathbf{X} = \{\mathbf{x}_1, \mathbf{x}_2, \ldots, \mathbf{x}_n\}$, distance metric $d$, stopping criterion $\epsilon$
2: **Output:** Clusters $C$
3: Initialize each point as its own cluster: $C = \{\{x_1\}, \{x_2\}, \ldots, \{x_n\}\}$
4: **while** the number of clusters $|C| > 1$ **do**
5:     Find the closest pair of clusters $C_i, C_j$ such that $d(C_i, C_j) = \min_{C_k, C_l \in C} d(C_k, C_l)$
6:     **if** $d(C_i, C_j) < \epsilon$ **then**
7:         Merge clusters: $C \leftarrow C \setminus \{C_i, C_j\} \cup \{C_i \cup C_j\}$
8:     **end if**
9: **end while**
10: **return** $C$

---

**Algorithm 2** ImputeINR Imputation Algorithm

---

1: **Input:** Time series data $\mathbf{X}$ with missing values, mask rate $r$
2: **Output:** Imputed values $\hat{\mathbf{X}}$
3: Perform feature clustering on the $N$ features of $\mathbf{X}$ to obtain clusters $C$
4: Reorder $\mathbf{X}$ based on clusters $C$ to get $\mathbf{X}'$
5: **for** each convolutional layer in the multi-scale feature extraction module **do**
6:     Extract features from $\mathbf{X}'$ using different kernel sizes
7: **end for**
8: Concatenate outputs from all convolutional layers to obtain $\dot{\mathbf{X}}$
9: Initialize INR tokens $\theta$
10: Input $\dot{\mathbf{X}}$ and $\theta$ into transformer encoder to predict the INR tokens $\theta^*$
11: **for** each timestamp $t$ **do**
12:     Query the corresponding value of the timestamp $t$ to get the imputed data $\hat{\mathbf{X}} = f_{\theta^*}(t)$
13: **end for**
14: **return** $\hat{\mathbf{X}}$

---

Based on this synthetic dataset, we test the representation capabilities of four different paradigms of INR continuous functions. As shown in the Figure 5, Model C demonstrates the fastest convergence speed and the best fitting results. This indicates that the representation capability of INR is strongest when both variable clustering and adaptive grouping are used simultaneously. In contrast, Model D has the worst fitting results, suggesting that the correlation information between variables from the same distribution significantly impacts the representation capability of INR. It is worth noting that in our ablation experiments, using variable clustering or adaptive grouping individually also improve the imputation ability. This is because the variable distributions in real datasets are more complex, making it challenging to separate variables belonging to the same cluster into different groups as in the synthetic dataset.

A.4    DETAILS OF BASELINE MODELS

The details of the baseline models are summarized here.

- **ImputeFormer**[1] A low-rank-induced Transformer that strikes a balance between strong inductive bias and high model expressiveness. By leveraging the inherent structures of spatiotemporal data, ImputeFormer learns well-balanced signal-noise representations, making it adaptable to a wide range of imputation challenges.

- **TimeMixer** [2] A fully MLP-based architecture incorporates Past-Decomposable-Mixing and Future-Multipredictor-Mixing blocks to effectively leverage disentangled multiscale series during both past extraction and future prediction phases.

- **iTransformer** [3] A transformer-based architecture which straightforwardly applies the attention mechanism and feed-forward network to the inverted dimensions. In this approach,

---

[1] https://github.com/tongnie/ImputeFormer
[2] https://github.com/kwuking/TimeMixer
[3] https://github.com/thuml/iTransformer

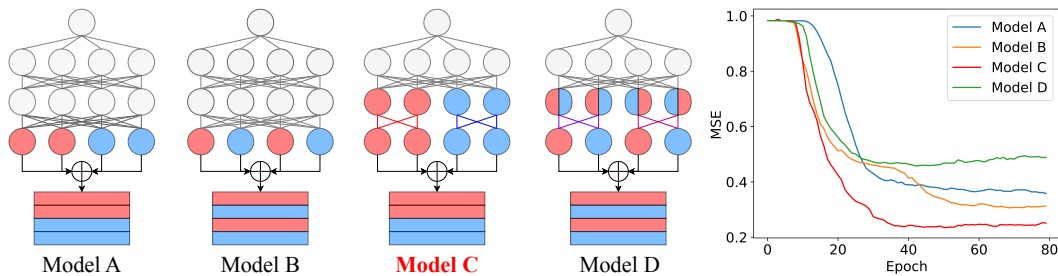

Figure 5: The four architectures we test to evaluate the representation capability of the INR continuous function for the synthetic time series dataset. The results prove that the representation capability of INR is strongest when both variable clustering and adaptive grouping are used simultaneously.

the time points of each individual series are embedded into variate tokens, which the attention mechanism uses to capture multivariate correlations. Simultaneously, the feed-forward network operates on each variate token to learn nonlinear representations.

- **FPT** [4] A frozen pre-trained transformer which leverages large language models on billions of tokens for time series analysis. Specifically, the self-attention and feedforward layers of the residual blocks in the pre-trained model are remained. It is assessed through fine-tuning across all major types of time series tasks.

- **TimesNet** [5] A method that transforms the 1D time series into a set of 2D tensors based on multiple periods. This transformation can embed the intraperiod- and interperiod-variations into the columns and rows of the 2D tensors respectively, making the 2D-variations to be easily modeled by 2D kernels.

- **SAITS** [6] A self-attention mechanism based method that learns missing values using a weighted combination of two diagonally-masked self-attention (DMSA) blocks. DMSA effectively captures both temporal dependencies and feature correlations across time steps, enhancing imputation accuracy and training speed. Additionally, the weighted combination allows SAITS to dynamically assign weights to the representations learned from the two DMSA blocks based on the attention map and missingness information.

- **BRITS** [7] A RNN-based method directly learns the missing values in a bidirectional recurrent dynamical system, without any specific assumption.

- **Transformer** It is the most basic transformer architecture, but in some cases, it performs better than other complex methods.

- **Mean/Median** It imputes missing entries using the mean or median values of the corresponding columns, yielding similar results for both methods.

---

[4]https://github.com/DAMO-DI-ML/NeurIPS2023-One-Fits-All/tree/main

[5]https://github.com/thuml/Time-Series-Library

[6]https://github.com/WenjieDu/SAITS

[7]https://github.com/caow13/BRITS