# OpenReview forum: "ImputeINR: Enhancing Time Series Imputation with Adaptive Group-based Implicit Neural Representations"
_ICLR.cc/2025/Conference — ICLR 2025 Conference Withdrawn Submission_

### Official Review · Reviewer_Hawo · 2024-10-31

**Soundness:** 3
**Presentation:** 3
**Contribution:** 3
**Rating:** 6
**Confidence:** 3

**Summary:**

The paper presents a novel approach for imputing missing values in time series data, particularly focusing on cases with high proportions of missing values. The proposed method, ImputeINR, employs implicit neural representations (INR) to model time series as continuous functions, allowing for fine-grained interpolation even with sparse observations. ImputeINR incorporates a multi-scale feature extraction module to capture various temporal patterns and a novel adaptive group-based architecture that leverages clustering to group variables with similar distributions. Finally, numerical experiments across seven datasets and five different levels of missing data were conducted to demonstrate the performance of  ImputeINR. Overall, the paper looks promising and organized, but I have some questions.

**Strengths:**

1. The author introduces an innovative ImputeINR model, addressing the complex problem of time series imputation under high missing ratios.  The method shows strong performance on benchmark datasets with diverse characteristics, achieving state-of-the-art imputation accuracy, especially under extreme mask rates.
2.  The paper includes extensive experiments, ablation studies, and robustness analyses to validate the contributions of each component, showing ImputeINR’s consistency across diverse settings.
3.  The paper provides clear explanations and visualizations, particularly for clustering results, aiding the reader's understanding of ImputeINR’s inner workings.

**Weaknesses:**

Please refer to the questions sections.

**Questions:**

1. In the Introduction, the authors state that ImputeINR is the first method designed specifically to handle extremely sparse observed data. However, to my knowledge, other recent studies have addressed this issue, including CSDI (Conditional Score-based Diffusion Models for Probabilistic Time Series Imputation) and SSSD (Diffusion-based Time Series Imputation and Forecasting with Structured State Space Models). The author should include these methods in their experimental comparisons or a comparative discussion of these approaches in the literature review and highlight how ImputeINR are different and better from these approaches for extremely sparse data scenarios.

2. Could the author provide recent advancements in INR for time series, limitations of current INR approaches, or how INR addresses missing values in existing work? For instance, to my knowledge, "Time Series Continuous Modeling for Imputation and Forecasting with Implicit Neural Representations" also provide an INR framework for data imputation, for which this is not being reviewed in this work.

3.  It is not so clear to me how the author generate the (missing) mask. Could the author clarify how their masks are generated in detail? In Section 4.1, the authors mention that missing values are generated by randomly masking values based on a specified mask rate. Does this imply the missing value are generated according to the missing rate? If so, have the authors also considered block missing scenarios, as proposed in SSSD, where entire segments are missing?

4. Could the authors clarify why the last column in Table 2, showing imputation results with the mean/median baseline, are identical?
Is this an error in reporting or is it a characteristic of the mean/median imputation method. If this is not an error, the authors could explain and discuss this pattern.

5. Additional detail on the computational complexity of ImputeINR would be helpful. Specifically, how does its computational burden compare to that of other baseline methods?

---

### Official Review · Reviewer_6xCc · 2024-11-01

**Soundness:** 1
**Presentation:** 2
**Contribution:** 2
**Rating:** 3
**Confidence:** 4

**Summary:**

This paper introduces an imputation approach to address the incomplete time series with high missing rates,. By leveraging implicit neural representations (INR) to learn continuous functions, ImputeINR can generate fine-grained imputations even when substantial values are missing. The method includes a multi-scale feature extraction module to enhance the imputation's fine-grained and global consistency. Additionally, ImputeINR uses a specific form of INR continuous function to separately learn trend, seasonal, and residual information, and an adaptive group-based framework to model complex residual information. Experiments on seven datasets show ImputeINR's superior performance in high absent ratios in time series.

**Strengths:**

S1. Imputing time series data is an important problem.

S2. Various baselines are considered in experiments.

S3. The visual analysis is conducted, to improve the readability.

**Weaknesses:**

W1. This paper focuses on imputing time series data with high missing rates, e.g., 70%, 90%. However, it is doubted whether this scenario is important and commonly observed in real applications, since there is no real scenario provided to support such a motivation. Please provide specific examples of real-world scenarios where such high missing rates occur, or to justify why addressing these extreme cases is important even if they are rare.

W2. The core idea is to use the ability of INR to learn continuous functions and achieve interpolation. To adapt INR for time series imputation, trend, seasonal and residual items are considered, which are typical operations for modeling temporal data. It is suggested to further highlight the contribution and novelty of proposed framework. Please explicitly compare the proposed approach to existing methods that use trend, seasonal, and residual decomposition, and clearly state what specific innovations this method introduces beyond these typical operations.

W3. As claimed in W1, in experiments, all the datasets are originally complete and only artificial missing values are considered in the evaluation. It is necessary to use real-world incomplete data with large missing rates in experiments to demonstrate the applicability of proposed methods as well as the motivation scenario.

W4. It is also suggested to consider the application study using real-world incomplete datasets, to investigate the performance of using proposed techniques to serve real scenarios.

W5. In Table 2, mean and median values are the same in different mask rates. Please explain why the mean and median values are the same across different mask rates, or to verify if this is correct.

W6. In Table 2, Mask rate is represented inconsistently, e.g., 10% or 0.1.

**Questions:**

Please see the weak points listed above.

---

### Official Review · Reviewer_MMwF · 2024-11-03

**Soundness:** 2
**Presentation:** 3
**Contribution:** 2
**Rating:** 3
**Confidence:** 5

**Summary:**

This paper presents a solution to the time series imputation problem, particularly for datasets with high proportions of missing observed values. The approach leverages Implicit Neural Representations (INR), which are known for their ability to model continuous functions, potentially enhancing the accuracy of missing data interpolation. The authors introduce three distinct functions to represent multivariate discrete datasets, corresponding to the trend, seasonal, and residual components. Additionally, a clustering module is incorporated to group channels with similar distributions, further refining the handling of inter-variable relationships and improving imputation quality. Experimental validation demonstrates the method's superior performance compared to existing techniques, particularly in extreme missing data scenarios.

**Strengths:**

1. The paper is well-written.
2. The targeted problem is important.

**Weaknesses:**

1. The word "novel" appears multiple times throughout the paper; however, techniques such as INR, clustering, and time series decomposition (including trend, seasonal, and residual components) are well-established and widely utilized in existing research. Additionally, the approach of learning parameters to calculate imputation results through mathematical methods has been explored in prior studies[1]. Although the combination of these elements may provide certain technical contributions, the work lacks a distinctive level of innovation to set it apart from previous works.
[1] Liu, Shuai, et al. "Multivariate time-series imputation with disentangled temporal representations." The Eleventh international conference on learning representations. 2023.

2. In Equations 9 and 10, the authors use fixed formulas to represent parts of the INR function (trend and seasonal components). While this choice enhances the interpretability of ImputeINR, it also reduces its ability to handle various complex datasets. Could the authors explain why they used these two fixed functions to represent the trend and seasonal components of time series data? Detailed insights will be preferred.

3. The average experimental results obtained by ImputeINR are quite impressive; however, this performance is primarily evident in datasets such as BAQ and IAQ, while the improvements observed on widely used datasets like ETT and Weather are quite modest. Does this indicate that ImputeINR has stringent requirements concerning dataset distribution? The authors are encouraged to clarify which types of datasets are most appropriate for imputation with ImputeINR.

4. The experimental settings require more detailed clarification. BAQ, IAQ, and Solar are datasets containing tens of thousands of time steps, while the experiments presented in this paper utilize only a few hundred or fewer. Furthermore, the training and testing set split ratios vary across datasets, such as Weather and Solar. Similar concerns are observed in other datasets as well.

5. Incorporating SOTA baselines, such as CSDI[2], would enhance the credibility of the experimental results.
[2] Tashiro, Yusuke, et al. "Csdi: Conditional score-based diffusion models for probabilistic time series imputation." Advances in Neural Information Processing Systems 34 (2021): 24804-24816.

**Questions:**

As seen in weaknesses.

---

### Official Review · Reviewer_vBf7 · 2024-11-06

**Soundness:** 3
**Presentation:** 3
**Contribution:** 3
**Rating:** 5
**Confidence:** 4

**Summary:**

This article proposes a new model for the problem of time-series imputation that uses Implicit Neural Representations (INR) for time-series modeling. The model employs a convolutional network to extract multi-scale features from the time series, followed by a transformer encoder. The time series are modeled by an INR decomposed into three sub-functions modeling the trend, seasonality, and residuals. The parameterization of the INR is performed based on the output of the transformer. To facilitate the learning of the residual function, clustering is performed on the different variables of the time series to dedicate a specific MLP to each cluster. Experiments on 7 datasets are conducted, comparing the proposed approach to various state-of-the-art algorithms. An ablation study is also carried out to demonstrate the usefulness of each component.

**Strengths:**

* The paper is well-written, the ideas are easy to follow, and the architecture is well described.
* The idea of using INRs for time series, and especially for imputation tasks, is a timely and important topic for the community.
* The results of the experiments are convincing, especially when the missing rate is high.

**Weaknesses:**

The greatest weakness of the paper, in my opinion, lies in the experimental section and the comparison to existing approaches. There are at least two other papers using INRs for time-series imputation: [1], which is also set in the context of extremely missing observed data (95%), and [2], which uses trend/seasonality/residual decomposition and is also set in the context of a high missing rate. The paper does not cite or compare to these two approaches. Given the similarity of ideas and the relatively few papers on the subject, it seems necessary for the proposed model to be compared to these two approaches.

A minor weakness is the section on clustering, which seems questionable to me. The assumption is that when variables are related, they remain related over time – as far as I understand, the clustering is done on the entire series. This clustering is only used in the residual part to facilitate learning. I wonder if the same result could not be achieved with better regularization of the MLP network used.

[1] Time Series Continuous Modeling for Imputation and Forecasting with Implicit Neural Representations, Le Naour et al., TMLR 2024
[2] HyperTime: Implicit Neural Representation for Time Series, E. Fons, A. Sztrajman, Y. El-Laham, A. Iosifidis, S. Vyetrenko, NeurIPS 2022 SyntheticData4ML

**Questions:**

Please see above.

---

### Author Response · Authors · 2024-11-23
**General Response**

We appreciate the time and effort invested by the reviewers in evaluating our work. The reviewers raised two main concerns: the discussion of existing INR-based time series representation methods and the comparison with current imputation methods designed for sparsely observed data. We will add the discussion in the revised version of the paper.

Given the relatively low initial scores, we decide to withdraw the submission at this stage.

Best regards,

Authors of submission 9758

---

### Note · Authors · 2024-11-23

I have read and agree with the venue's withdrawal policy on behalf of myself and my co-authors.